# Multicalibration Yields Better Matchings

**Riccardo Colini Baldeschi** [1]  **Simone Di Gregorio** [2]  **Simone Fioravanti** [2]  **Federico Fusco** [2]  **Ido Guy** [1]
**Daniel Haimovich** [1]
**Stefano Leonardi** [2]  **Fridolin Linder** [1]  **Lorenzo Perini** [1]  **Matteo Russo** [3]  **Cem Sirin** [2]  **Niek Tax** [1]

## Abstract

Consider the problem of finding the best matching in a weighted graph where we only have access to predictions of the actual stochastic weights, based on an underlying context. If the predictor is the Bayes optimal one, then computing the best matching based on the predicted weights is optimal. However, in practice, this perfect information scenario is not realistic. Given an imperfect predictor, a suboptimal decision rule may compensate for the induced error and thus outperform the standard optimal rule. In this paper, we propose *multicalibration* as a way to address this problem. This fairness notion requires a predictor to be unbiased on each element of a family of protected sets of contexts. Given a class of matching algorithms $\mathcal{C}$ and any predictor $\gamma$ of the edge-weights, we show how to construct a specific multicalibrated predictor $\hat{\gamma}$, with the following property. Picking the best matching based on the output of $\hat{\gamma}$ is competitive with the best decision rule in $\mathcal{C}$ applied onto the original predictor $\gamma$. We complement this result by providing sample complexity bounds, and by performing numerical experiments.

## 1. Introduction

The interplay of classical algorithms and machine learning routines in industry pipelines is a well-established phenomenon: optimization algorithms are used to *fit* learning models, which, in turn, are used to *generate* inputs for algorithmic tasks, or to *guide* them. While one side of this synergy is well understood, only recently has the theory community started investigating how machine learning may actually help to answer classical algorithmic questions.

In the *algorithms with predictions* framework (Mitzenmacher & Vassilvitskii, 2020; Balkanski et al., 2023), the goal is to investigate the extent to which additional information provided by some machine learning prediction can improve the worst-case theoretical guarantees. If such additional information is correct, then the performance should improve, while the whole pipeline should be robust with respect to bad quality predictions. Similarly, in *Data-Augmented Algorithm Design* (Balcan, 2020), the focus is on learning from data the best algorithm on a specific input distribution. In this paper, we investigate a problem that is similar in spirit to these lines of work and is practically motivated. Imagine running an optimization task on an input that is not fully known, but whose relevant features are only predicted by some machine learning black box. Our goal is to understand how such a prediction can be modified *ex-post*, to improve the overall quality of the algorithmic solution.

As a first example, consider the problem of choosing the best out of $d$ actions. The environment is modeled by a pair of random vectors $(V, X) \sim \mathcal{D}$, where $X \in \mathbb{R}^n$ represents context/features that can be used to guide the decision, and $V \in [0, 1]^d$ contains the actions' rewards. If we know $\mathcal{D}$, then the value-maximizing strategy entails choosing $\arg\max \mathbb{E}[V_i|X]$; however, in applications, we can only base our decision on a predictor $\gamma : \mathbb{R}^n \to [0, 1]^d$ of such quantity, generated by some black-box machine learning routine. Clearly, we are free to apply many decision rules to $\gamma(X) \in [0, 1]^d$, the most natural one being the $\arg\max$. However, it is fairly easy to construct examples where even unbiased estimators may perform arbitrarily badly.

To illustrate this point, let $\varepsilon$ be an arbitrarily small parameter, and consider two deterministic arms, one with value $1$ and the other with value $1/\varepsilon$, while the context $X$ is drawn independently and uniformly in $[0, 1]$. Denote with $\gamma$ the estimator that is always correct on the first action, but outputs $1/\varepsilon^2$ for the second arm if $X \in [0, \varepsilon]$ and $0$ otherwise. By definition, $\gamma$ estimates the first arm perfectly; for the second arm (whose value is deterministically $1/\varepsilon$), $\gamma$ has value $0$ with probability $1 - \varepsilon$ and $1/\varepsilon^2$ with the remaining proba-

[1]Meta Central Applied Science, UK [2]Dept. of Computer, Control and Management Engineering, Sapienza University of Rome, Rome, Italy [3]EPFL, Switzerland. Correspondence to: Simone Di Gregorio <simone.digregorio@uniroma1.it>.

*Proceedings of the 43rd International Conference on Machine Learning*, Seoul, South Korea. PMLR 306, 2026. Copyright 2026 by the author(s).

bility. Consequently, the expected value of the predictor for the second arm is exactly $1/\varepsilon$, so that the predictor $\gamma$ is indeed unbiased. Now observe that the second arm strictly dominates the first, with the gap in utility between the two arms growing arbitrarily large as $\varepsilon \to 0$. However, making a choice trusting the predictor's values results in choosing the suboptimal first arm with probability $1 - \varepsilon$. In particular, the naive decision rule "always choose the second action" strictly dominates the apparently optimal one of taking the action with largest predicted value.

Given a family of candidate decision rules $\mathcal{C}$, and a black-box predictor $\gamma$, we want to *multicalibrate* $\gamma$ in such a way that the $\arg\max$ over the new predictor performs at least as well as the best decision rule in $\mathcal{C}$ over the initial predictor $\gamma$. Note, we do not know the underlying distribution, nor how the predictor $\gamma$ is actually computed, but we would like to modify the predictor in such a way that simply feeding its output in an optimization routine would perform as well as the best decision rule for $\gamma$ within a given class $\mathcal{C}$.

## 1.1. Our Results

Our contribution in this direction is conceptual: we argue that the right property for a predictor to have in this setting is a version of *multicalibration* (Hébert-Johnson et al., 2018), a notion of calibration stemming from the literature on algorithmic fairness. In words, multicalibration requires a predictor to be calibrated on each element of a family of protected sets of contexts. In this application, such a family depends on the structure of the problem at hand and the class of matching algorithms $\mathcal{C}$. In particular, instead of requiring an estimator to be unbiased over protected sets, we use multicalibration to "protect" events of the type "a given edge $e$ is chosen by a certain decision $c \in \mathcal{C}$". Indeed, for the sake of generality, we consider the natural optimization task of finding a max-weight matching in an $n$-node graph where the edge-weights are stochastic, and we only have access to them via a black-box predictor $\gamma$. Given a finite class of matching algorithms $\mathcal{C}$, in Theorem 3.1 we show that a suitably multicalibrated predictor $\hat{\gamma}$ exists such that computing the $\arg\max$ matching on the edges predicted by $\hat{\gamma}$ is optimal, up to an additive precision $\varepsilon$. This predictor can be constructed efficiently, with $\tilde{O}\left(n^{3.5}/\varepsilon^3 \log |\mathcal{C}|\right)$* many samples from $\mathcal{D}$ using techniques from adaptive data analysis (Dwork et al., 2015), or with $\tilde{O}\left(n^5/\varepsilon^4 \log |\mathcal{C}|\right)$ many samples when implementing boosting by iterating over disjoint parts of the dataset.

An alternative to our approach would consist in estimating the expected performance of each algorithm sampling from $\mathcal{D}$, and then committing to the best one. While this would require fewer samples (i.e., $\tilde{O}(n^2/\varepsilon^2 \log |\mathcal{C}|)$), our modular result enjoys two desirable properties:

(i) we improve on the original black-box predictor, both for the task at hand and in mean square error sense.

(ii) we can decide ex-ante what the best algorithm will look like, as long as it is optimal with perfect information, meaning that we can reuse the same one for different learning instances.

Stated differently, we only need to consider the matching algorithms in $\mathcal{C}$ during the preprocessing phase (in which we multicalibrate the predictor); afterwards, we only need to find the max-weight matching on the adjusted predictor. This means that in the "test phase" we only need one algorithm. The procedure is illustrated in Figure 1.

To see why this is relevant, consider the lifecycle of a real-world, large-scale matching system operating with unknown inputs. The standard design pattern is "predict-then-optimize": feeding predicted weights into a standard matching algorithm (e.g., max-weight matching). When developers inevitably identify scenarios where this pipeline makes suboptimal decisions, the temptation is to intervene with manual patches—ad-hoc "IF-THEN" rules that override the solver in specific contexts. While this may be an immediate fix, it introduces significant system complexity and "unlearned" logic that becomes technical debt. If the data distribution shifts, these hard-coded patches may degrade performance and require manual removal. Multicalibration offers a rigorous alternative to this patching cycle. Instead of hard-coding a heuristic into the decision logic, the developer can simply include the heuristic as a test function in the multicalibration procedure. This guarantees that the standard system remains competitive with the heuristic, preserving architectural simplicity downstream while ensuring the decision logic remains data-driven and adaptive.

Finally, we stress that our sample complexity bound is worst-case with respect to the initial predictor $\gamma$. Since the analysis is based on a potential argument that uses the expected mean square error (MSE) as potential, the closer the original predictor is to the Bayes optimal, the fewer samples are needed to multicalibrate. This is realistic for the quality of the predictors used in industry. For further details, in Theorem 3.3, we provide a refined sample-complexity analysis which depends explicitly on the quality of the initial predictor.

**Beyond Matching.** Our choice of max-weight matching as a running example is just for ease of presentation: our approach is flexible and applies easily to any linear maximization task with deterministic constraints (as in e.g., finding the max-weight independent set in a matroid or learning with rejection). Similarly, with proper adjustments, our analysis goes through even if we cannot solve optimally the underlying problem, but have an approximation routine.

---

*The notation $\tilde{O}$ hides terms that are polylogarithmic in $n$, $1/\varepsilon$.

*Figure 1.* Proposed pipeline to obtain a matching: (1) the initial predictor $\gamma$ is post-processed to a multicalibrated $\hat{\gamma}$, (2) the final matching is chosen according to $c^\star$.

## 1.2. Related Work

Multicalibration, originally introduced by Hébert-Johnson et al. (2018), has garnered significant attention due to its versatility. Beyond the authors' initial analysis of sample complexity and its relationship to boosting, the concept has inspired new notions such as *omnipredictors* (Gopalan et al., 2022a) and *outcome indistinguishability* (Dwork et al., 2021; Gopalan et al., 2023), while demonstrating deep connections to complexity theory (Casacuberta et al., 2024). Subsequent research has proposed various extensions. Jung et al. (2021) introduce multicalibration requirements for higher moments, while Haghtalab et al. (2023) situate the concept within multi-objective learning. Most relevant to our work is the framework of *weighted* multicalibration proposed by Gopalan et al. (2022b). Their approach utilizes two function classes—a hypothesis class for protected sets and a weight class to adjust probabilistic requirements. We adapt their general multiclass classification framework to define our choice of multicalibration for vector-valued functions. Similarly, Blasiok et al. (2024) introduce multicalibration on *auditing* functions, which depend on both inputs and predictor outputs—an approach closely aligned with our setting. Finally Hu et al. (2023) adapt the omnipredictor definition to tackle constrained loss minimization.

Zhao et al. (2021) investigates decision-making using predicted labels when an initial multi-class classification predictor is post-processed and made appropriately calibrated. They prove that in such scenario the arg max is then the only rational choice on the resulting predictor. Although their high level motivation is affine to ours, their result are orthogonal to ours, as they do not provide any guarantees that enable a comparison with the decision-making performance of the original predictor when using a generic policy within a given family (which is our benchmark).

In independent and concurrent work, Kiyani et al. (2026) build upon Zhao et al. (2021) to investigate a similar setting, when the underlying predictor is calibrated against a generic family $\mathcal{H}$. Under specific assumptions, they derive an explicit formula for modifying predicted labels ex-post,

ensuring that the arg max (with respect to these modified predictors) is optimal in a minimax sense. They further identify conditions on $\mathcal{H}$ for the direct arg max of the predicted labels to be optimal.

In contrast, we define protected groups based on an input predictor $\gamma$. By leveraging multicalibration, we guarantee performance competitive with the best possible post-processing of $\gamma$, effectively capturing all decision-relevant signals present in the initial model. While robust decision-making often necessitates conservative actions to mitigate worst-case uncertainty, our boosting approach actively refines the model to recover the optimal matching whenever the original predictor contains sufficient information.

## 2. Preliminaries

**Multicalibration.** Let $\mathcal{X}$ be a feature space and $\mathcal{Y} = [0, 1]^d$ the label space. We consider point-label pairs $(x, y)$ sampled from an unknown probability distribution $\mathcal{D}$ supported on $\mathcal{X} \times \mathcal{Y}$. Given any predictor $f : \mathcal{X} \to \mathcal{Y}$, *multicalibration* (Hébert-Johnson et al., 2018) requires $f$ to well approximate $\mathbb{E}[y|x]$ on average in each set of a given family of *protected* subsets of $\mathcal{X}$. To make it feasible to check this condition in practice, these sets must have some structure (i.e. finite Vapnik-Chervonenkis dimension) or be finite. Additionally, the guarantee should degrade with the probability of these sets, to account for low sample frequency.

The definition of multicalibration we use in this paper is very general, and comes from adapting to regression the one from Gopalan et al. (2022b), which is specialized to classification[†]. More specifically, notice that the notion of protected sets — normally induced by the hypothesis class — is in our case absorbed into the weights themselves, that are now allowed to depend directly on $x$.

**Definition 2.1** (Weighted Multicalibrated predictor). Given a weight class $\mathcal{W} \subseteq \{[0, 1]^d \times \mathcal{X} \to \{0, 1\}^d\}$ and $\alpha \geq 0$, a

---

[†]Boosting-like approaches to attain multicalibration are not impacted by the switch in the setting. What changes is that we need to scale when using the same analysis.

predictor $f : \mathcal{X} \to [0,1]^d$ is $(\mathcal{W}, \alpha)$-multicalibrated if for every $w \in \mathcal{W}$ we have:

$$|\mathbb{E}_{\mathcal{D}} \left[ \langle w(f(x), x), y - f(x) \rangle \right]| \leq \sqrt{d} \cdot \alpha. \qquad (1)$$

We highlight that the expectation in the definition is taken with respect to the distribution $\mathcal{D}$, thus formalizing the intuition that $f$ should be accurate on average, when summing over the contributions from the sets induced by each entry of a $w \in \mathcal{W}$. Indeed, in our setting, by the tower property of conditional expectations, the expectation in Equation (1) does not change if $y$ is replaced by the Bayes predictor $\mathbb{E}\left[ y | x \right]$. We observe that the presence of indicator terms (the weights) inside the expectation in Equation (1) allows for this condition to be practically relevant, as it is robust against sets of low probability.

## 3. Learning Matchings with Multicalibrated Predictions

In this section, we present our results in the maximum matching framework. We consider a complete undirected graph $G = (V, E)$ and denote with $n$ and $m$ the number of nodes and edges, respectively. We allow the graph to have null weights on the edges, so the assumption of completeness is without loss of generality.

### 3.1. Optimization Guarantees

Let $\mathcal{C}$ be a finite and fixed family of algorithms for max-weight matching, and denote with $c^*$ the optimal algorithm for such a combinatorial task (for instance, Edmonds algorithm—see Chapter 26 in (Schrijver, 2003)). We investigate a statistical scenario where $(x, y) \in \mathcal{X} \times [0,1]^m$ are sampled i.i.d. from a joint distribution $\mathcal{D}$. The set $\mathcal{X}$ is a generic space of context, while $y_e$ for $e \in E$ represents the random weight of the edge. We are also given a generic predictor $\gamma : \mathcal{X} \to [0,1]^m$, such that $\gamma_e(x)$ is a prediction for $y_e$. Note, we do not make any assumption on $\gamma$.

We want to build a new predictor $\hat{\gamma}$ that enjoys the following property:

$$\max_{c \in \mathcal{C}} \mathbb{E} \left[ \sum_{e \in M_c(\gamma(x))} y_e \right] \lesssim \mathbb{E} \left[ \sum_{e \in M_{c^\star}(\hat{\gamma}(x))} y_e \right], \qquad (2)$$

where $M_c(\gamma)$ and $M_c(\hat{\gamma})$ denote the matchings output by decision rule $c$ on input weights $\gamma$ and $\hat{\gamma}$ respectively.

Specifically, we define $\hat{\gamma}$ as a $(\mathcal{W}, \alpha)$-multicalibrated predictor with respect to a weight class $\mathcal{W}$, constructed as follows.

We introduce $\mathcal{W} = \mathcal{W}_1 \cup \{w^\star\}$, where:

$$\begin{aligned}
w_c(p, x) &= (\mathbb{1}_{\{e \in M_c(\gamma(x))\}})_e \\
\mathcal{W}_1 &= \{w_c\}_{c \in \mathcal{C}} \\
w^\star(p, x) &= (\mathbb{1}_{\{e \in M_{c^\star}(p)\}})_{e \in E}
\end{aligned} \qquad (3)$$

Each indicator function in this set essentially filters edges based on predictions or contexts, extracting the associated errors. Notice that the $w_c$'s are constant in the first argument, while the value of $w^\star$ is not affected by the second argument. Intuitively, this construction mirrors the two steps of our theoretical analysis. The set $\mathcal{W}_1$ ensures that our new predictor $\hat{\gamma}$ faithfully captures the value of the algorithms in $\mathcal{C}$ that take as input the original predictor $\gamma$. Specifically, for any algorithm $c \in \mathcal{C}$, if the original predictor suggests a set of edges through $c$, $\hat{\gamma}$ must be unbiased on that set.

Conversely, $w^\star$ ensures *self-consistency*: it forces $\hat{\gamma}$ to be unbiased on the edges selected by the optimal algorithm $c^*$ when driven by $\hat{\gamma}$ itself. By satisfying both conditions simultaneously, the multicalibrated predictor allows us to compare the performance of any of the original decisions on $\gamma$ against the standard optimal policy on $\hat{\gamma}$.

We claim that if $\hat{\gamma}$ is a multicalibrated predictor w.r.t. $\mathcal{W}$, using $c^\star$ in the predictions given by $\hat{\gamma}(x)$ is on average (up to an additive constant depending on the $\alpha$ from Equation (1)) as good as taking the best $c \in \mathcal{C}$ according to $\gamma$.

**Theorem 3.1.** *Let $\varepsilon \in (0,1)$ be any fixed precision and set the multicalibration parameter $\alpha = \varepsilon/2\sqrt{m}$. If $\hat{\gamma}$ is $(\mathcal{W}, \alpha)$-multicalibrated, it holds that:*

$$\max_{c \in \mathcal{C}} \mathbb{E} \left[ \sum_{e \in M_c(\gamma(x))} y_e \right] \leq \varepsilon + \mathbb{E} \left[ \sum_{e \in M_{c^\star}(\hat{\gamma}(x))} y_e \right].$$

*Proof.* We suppress $x$ as argument to the predictors, to simplify the notation. In the following, $c$ is a generic matching function in $\mathcal{C}$. We start by arguing that the new predictor $\hat{\gamma}$ well estimates the weight of the matching $M_c(\gamma)$, for any $c \in \mathcal{C}$. We have the following:

$$\mathbb{E} \left[ \sum_e (y_e - \hat{\gamma}_e) \mathbb{1}_{\{e \in M_c(\gamma)\}} \right]$$

$$= \mathbb{E} \left[ \langle \underbrace{(\mathbb{1}_{\{e \in M_c(\gamma)\}})_e}_{w \in \mathcal{W}_1 \subset \mathcal{W}}, y - \hat{\gamma} \rangle \right] \leq \alpha \sqrt{m} \qquad (4)$$

The inner product appearing above is indeed upper-bounded by $\alpha \cdot \sqrt{m}$ due to our notion of multicalibration and our choice of $\mathcal{W}_1$. We now relate the expected weight of $M_c(\gamma)$ with respect to $\hat{\gamma}$ with the actual expected weight of $M_{c^*}(\hat{\gamma})$ (as measured by $y_e$).

$$\mathbb{E}\left[\sum_e \hat{\gamma}_e \mathbb{1}_{\{e \in M_c(\gamma)\}}\right]$$

$$\leq \mathbb{E}\left[\sum_e \hat{\gamma}_e \mathbb{1}_{\{e \in M_{c^\star}(\hat{\gamma})\}}\right] \qquad (c^\star(\hat{\gamma}) \text{ optimal for } \hat{\gamma})$$

$$= \mathbb{E}\left[\sum_e (\hat{\gamma}_e - y_e)\mathbb{1}_{\{e \in M_{c^\star}(\hat{\gamma})\}}\right] + \mathbb{E}\left[\sum_e y_e \mathbb{1}_{\{e \in M_{c^\star}(\hat{\gamma})\}}\right]$$

$$= \mathbb{E}\left[\langle \underbrace{(\mathbb{1}_{\{e \in M_{c^\star}(\hat{\gamma})\}})_e}_{w^\star \in \mathcal{W}}, \hat{\gamma} - y\rangle\right] + \mathbb{E}\left[\sum_e y_e \mathbb{1}_{\{e \in M_{c^\star}(\hat{\gamma})\}}\right]$$

$$\leq \alpha \cdot \sqrt{m} + \mathbb{E}\left[\sum_{e \in M_{c^\star}(\hat{\gamma})} y_e\right].$$

Note, the last inequality follows by our multicalibration setup. Plugging in the above inequality in Equation (4) yields the following:

$$\mathbb{E}\left[\sum_{e \in M_c(\gamma)} y_e\right] - \mathbb{E}\left[\sum_{e \in M_{c^\star}(\hat{\gamma})} y_e\right] \leq 2\alpha\sqrt{m}$$

Since the choice of $c \in \mathcal{C}$ was arbitrary, the result holds when taking the maximum over $\mathcal{C}$. □

### 3.2. Iterative Boosting Algorithm

Algorithm 1 implements our notion of multicalibration in practice, given access to an i.i.d. dataset from the underlying distribution. This iterative boosting procedure initializes $\hat{\gamma}$ as the original $\gamma$ and then, at each iteration $t$: (1) it employs a CHECK function to identify violations of the multicalibration condition for $\hat{\gamma}^t$, returning the violating weight function and its sign, and (2) it performs a projected gradient descent step on $\hat{\gamma}^t$. The algorithm is adapted from the procedure described in Gopalan et al. (2022b). Note that the function $\text{proj}_{[0,1]^m}(x)$ in line 11 denotes the clipped vector whose component $i = 1, \ldots, m$ equals $\min(\max(x_i, 0), 1)$.

The CHECK routine serves as the core component of Algorithm 1. It is a learning procedure that takes as input a batch of $N$ i.i.d. data points and searches for a weight function $w \in \mathcal{W}$ such that the estimated average of $\langle w(\hat{\gamma}^t(\cdot), \cdot), y - \hat{\gamma}^t(\cdot)\rangle$ exceeds $\alpha \cdot \sqrt{m}$. Regardless of its implementation, it must satisfy the following properties.

**Requirements 3.2** (CHECK function). Fix $\alpha > 0$ and a predictor $\gamma$. Given a sample $D = \{(x^j, y^j)\}_{j=1}^N$, the routine $\text{CHECK}_{\mathcal{W}, \alpha, \gamma}$ either returns a pair $(w, b) \in \mathcal{W} \times \{-1, +1\}$ or the symbol $\perp$. Specifically:

1. If there exists $w' \in \mathcal{W}$ such that

$$\text{1}/\sqrt{m}\,\mathbb{E}_{\mathcal{D}}[\langle w'(\hat{\gamma}(x), x), y - \hat{\gamma}(x)\rangle] \notin [-\alpha, \alpha],$$

---

**Algorithm 1** WeightedMC$(\alpha, \mathcal{W}, \{(x^j, y^j)\}_{j=1}^N, \gamma)$

1: $\hat{\gamma}^0(\cdot) \leftarrow \gamma$
2: $\eta \leftarrow \alpha/2$
3: $t \leftarrow 0$
4: $D \leftarrow \{x^j, y^j\}_{j=1}^N$
5: **while** true **do**
6:    **if** $\text{CHECK}_{\mathcal{W}, \alpha, \hat{\gamma}^t}(D) = \perp$ **then**
7:       **break**
8:    **else**
9:       $w_{t+1}, b_{t+1} \leftarrow \text{CHECK}_{\mathcal{W}, \alpha, \hat{\gamma}^t}(D)$
10:       $\delta_{t+1}(\cdot) \leftarrow b_{t+1} \cdot w_{t+1}(\hat{\gamma}^t(\cdot), \cdot)$
11:       $\hat{\gamma}^{t+1}(\cdot) \leftarrow \text{proj}_{[0,1]^m}(\hat{\gamma}^t(\cdot) + \eta \cdot \delta_{t+1}(\cdot))$
12:       $t \leftarrow t + 1$
13:    **end if**
14: **end while**
15: **return** $\hat{\gamma}^t$

---

then $w \neq \perp$ and the following holds:

$$\text{1}/\sqrt{m}|\mathbb{E}_{\mathcal{D}}[\langle w(\hat{\gamma}(x), x), y - \hat{\gamma}(x)\rangle]| \geq \alpha/2 \text{ and}$$
$$b = \text{sign}\left(\mathbb{E}_{\mathcal{D}}[\langle w(\hat{\gamma}(x), x), y - \hat{\gamma}(x)\rangle]\right),$$

where $\text{sign}(x) = x/|x|$;

2. If $w = \perp$, then we have that for all $w' \in \mathcal{W}$, $\text{1}/\sqrt{m}\,\mathbb{E}_{\mathcal{D}}[\langle w'(\hat{\gamma}(x), x), y - \hat{\gamma}(x)\rangle] \in [-\alpha, \alpha]$.

### 3.3. Analysis of Algorithm 1

We now analyze the theoretical guarantees of the proposed algorithm. First observe that, upon convergence, the properties described in Requirements 3.2 ensure that the final output satisfies the multicalibration conditions. It remains to assess the algorithm's convergence and sample complexity. Our results depend on two factors: (1) the availability of an upper bound $r$ to the MSE of the initial predictor $\gamma$; (2) the implementation of the CHECK routine. Below, we focus on (1), deferring the discussion of how modifying CHECK improves sample complexity to the end of this section.

Convergence in a finite number of iterations is established via a potential argument and holds regardless of whether the estimate $r$ is available. Leveraging $r$ requires a more fine-grained analysis, which makes the upper bound on the number of iterations directly proportional to $r$ and leads to a more general result. Our main findings are summarized in Theorem 3.3. We note that the sample complexity bound in the theorem takes into account a statistical implementation of CHECK which utilizes a distinct subset of the data for each call.

**Theorem 3.3.** *Fix any failure probability $\delta \in (0, 1)$, precision $\varepsilon \in (0, 1)$, and initial predictor $\gamma$. Assume that $\gamma$ has mean square error $\mathbb{E}\left[\|\gamma - \mathbb{E}[y|x]\|_2^2\right] \leq r$ for some known*

*r. Then Algorithm 1 with $\alpha \in O(\varepsilon/n)$ enjoys the following properties:*

- *It converges in $O\left(r/n\alpha^2\right)$ iterations of the while loop;*

- *It requires $O\left(\frac{rn^3 \cdot \log\left(\frac{nr|\mathcal{C}|}{\varepsilon\delta}\right)}{\varepsilon^4}\right)$ i.i.d. samples from $\mathcal{D}$;*

- *With probability $(1-\delta)$ it returns a predictor $\hat{\gamma}$ s. t.*

$$\max_{c \in \mathcal{C}} \mathbb{E}\left[\sum_{e \in M_c(\gamma(x))} y_e\right] \le \varepsilon + \mathbb{E}\left[\sum_{e \in M_{c^\star}(\hat{\gamma}(x))} y_e\right].$$

*Proof.* As a first step, we prove the bound on the number of iterations required for convergence. We define the value of the potential $\phi$ at the $t^{\text{th}}$ iteration as

$$\phi(\hat{\gamma}^t) = \mathbb{E}\left[\|\hat{\gamma}^t - \mathbb{E}[y|x]\|_2^2\right] \quad (5)$$

By expanding this squared norm and using that the value of a matching is bounded by $\sqrt{m}$, we obtain that the decrease in potential between iteration $t$ and $t+1$ satisfies the following:

$$\phi(\hat{\gamma}^t) - \phi(\hat{\gamma}^{t+1}) \ge \alpha\sqrt{m}\eta - \eta^2\sqrt{m}$$
$$= \sqrt{m}\alpha^2/4. \quad \text{(As } \eta = \alpha/2\text{)}$$

Now, let $T$ be the number $T$ of iterations of the loop. To bound $T$ we need to solve the inequality $\phi(\hat{\gamma}^0) - T \cdot \sqrt{m}\alpha^2/4 \ge 0$. Since $\phi(\hat{\gamma}^0) = \phi(\gamma) = r$, the number of iterations linearly shrinks with $r$, and we get the first of our desired results.

Now we want to bound the sample complexity of statistically implementing the oracle call to CHECK. Let $N' \le N$ be the number of samples from $D$ consumed by CHECK in a single call. For any $w \in \mathcal{W}$, let $z_w^j = 1/\sqrt{m}\langle w(\hat{\gamma}^t(x^j), x^j), y^j - \hat{\gamma}^t(x^j)\rangle$, denoting its empirical average and expectation with $\hat{z}_w$ and $\bar{z}_w$, respectively. Clearly, $z_w^j \in [-1, 1] \;\forall j \in [N'], w \in \mathcal{W}$, due to our normalization by $\sqrt{m}$, again using that the value of a matching is bounded by $\sqrt{m}$. We claim that, with enough samples, the procedure that returns the index of any $\hat{z}_w \notin [-\alpha/2, \alpha/2]$ is a correct implementation of a call to CHECK, with probability $1 - \delta_0$, where $\delta_0$ is a failure parameter that we will fix later.

By Hoeffding's Inequality and a union bound, we have:

$$\mathbb{P}\left(\exists w \in \mathcal{W} : |\hat{z}_w - \bar{z}_w| \ge \alpha/2\right) \le 2|\mathcal{W}|e^{-1/8 N'\alpha^2} = \delta_0.$$

Therefore, setting $N' \in O(\log(|\mathcal{W}|/\delta_0)/\alpha^2)$ suffices to concentrate every $\hat{z}_w$ in an interval of length $\alpha$ around its mean $\bar{z}_w$, with probability $1 - \delta_0$. This implies that, with the same probability, if a single $\bar{z}_w \notin [-\alpha, \alpha]$, $\hat{z}_w \notin [\alpha/2, \alpha/2]$ and its weight $w$ will be returned by the procedure. This proves our claim.

Now, as already argued, we call CHECK at most $4r/\sqrt{m}\alpha^2$ times; therefore we need to union bound over the correctness of every call, meaning that we need to consider $\delta_0 = \sqrt{m}\alpha^2\delta/4r$. Multiplying the number of calls by the sample complexity of each call, we get a sample complexity:

$$N \in O\left(\frac{r\log\left(|\mathcal{W}|r/\sqrt{m}\alpha^2\delta\right)}{\sqrt{m}\alpha^4}\right).$$

Putting $\alpha = \varepsilon/2\sqrt{m}$ as required in Theorem 3.1, we get the claimed result. □

**Discussion of the sample complexity.** Theorem 3.3 implies that the worst-case sample complexity (i.e. the one obtained with $r \in O(n^2)$) is $\tilde{O}(n^5/\varepsilon^4)$. Realistically, one can expect that predictors (like $\gamma$ in our setting) used in the industry attain a reasonable small mean-squared error. Consequently, if the target precision in Theorems 3.1 and 3.3 is $\varepsilon$, it is natural to require the mean-squared error on each edge to be $O(\varepsilon^2)$. Under this realistic assumption, the sample complexity reduces to $\tilde{O}(n^5/\varepsilon^2)$.

We mention that the worst-case dependency on $\varepsilon$ can be improved to $1/\varepsilon^3$ (up to additional polylogs) by resorting to adaptive data analysis techniques (Dwork et al., 2015), where the CHECK query is not implemented by considering disjoint subsets of the data, but by repeatedly evaluating a randomized implementation of the query on the same data. Clearly, this induces dependencies between different iterations, and thus we say these queries are adaptively chosen. As for Gopalan et al. (2022b), it is possible to cast CHECK as a routine returning the maximizer of the calibration error over $\mathcal{W}$ and use an immediate adaptation of Corollary 6.4 from Bassily et al. (2021). The analysis is then the same as the proof of Theorem 3.3, just with a more sample efficient implementation of CHECK. Considering thus again the bound on the number of iterations[‡], we obtain the following sample complexity bound:

$$O\left(\frac{n^{2.5}\sqrt{r}\log(n|\mathcal{C}|/\varepsilon)\log^{3/2}(n/\varepsilon\delta))}{\varepsilon^3}\right).$$

This implementation relies on the exponential mechanism (McSherry & Talwar, 2007) over $\mathcal{W}$, ensuring that the $O(N|\mathcal{W}|)$ iteration running time remains comparable to the one of Algorithm 1 with a simpler implementation of CHECK, which requires $O(N'|\mathcal{W}|)$ time per iteration.

# 4. Other Models

In this Section, we briefly explain how to generalize our approach to other linear optimization problems.

---

[‡]Now, every iteration is a query adaptively chosen based on the past ones.

**Finding the best action.** This is the problem discussed in the introduction: there are $m$ actions, each characterized by a value $y_i$, and the learner has access to a context vector $x \in \mathcal{X}$ that is drawn from a joint distribution $\mathcal{D}$ over $\mathcal{X} \times [0,1]^m$. Since choosing the best action is equivalent to finding the max-weight matching in a star graph of $m + 1$ nodes (a central node linked to the $m$ action nodes), our results for matching do carry over immediately, with an improved (worst-case) sample complexity of $\tilde{O}(\sqrt{m}/\varepsilon^3 \log |\mathcal{C}|)$ since only one edge is active for every decision output and we thus do not need to scale down the multicalibration error $\alpha$ as we do for matching. Note that this model captures the standard classification task, modulo projecting to the simplex instead than to $[0,1]^m$.

**Learning with Rejection.** Consider now the supervised learning framework with *rejection* (Hendrickx et al., 2024), where the learner can either predict the label of the example seen, or "reject" it. Typically, the cost of rejecting a point is smaller than misclassifying it. If the underlying learning task is binary classification, this can be embedded in the best-action framework by assuming the existence of three actions: "predict YES", "predict NO", and "reject". The predictor then correspondingly has three outputs, the first two modeling conditional probabilities and the third fixed to a specific value, based on the cost of rejecting; this third output is never updated, while the first two are projected back to the 1-dimensional simplex after every update, if needed. Since we only have a constant number of actions, we only need $\tilde{O}(1/\varepsilon^3 \log |\mathcal{C}|)$ samples.

**Max-Weight Base.** The basic setting with $n$ elements, each characterized by a random value $y_i \in [0,1]$ is similar to the ones above, with the difference that (i) a matroid[§] is defined on the elements and (ii) the algorithm designer can select any subset $S$ of elements that is independent with respect to the matroid. Let r be the rank of the matroid[¶], then we can carry over the same analysis as in matching, with the difference that Equation (4) is upper bounded by $\alpha \cdot$ r. This implies that setting $\alpha = \varepsilon/\text{r}$ is enough to get the same approximation result, for an overall sample complexity of $\tilde{O}(\sqrt{n} \cdot \text{r}^{2.5}/\varepsilon^3 \log |\mathcal{C}|)$.

## 5. Empirical Evaluation

In this section, we empirically evaluate the proposed method in two synthetic settings characterized by model misspeci-

fication: *Finding the best action*, or *Best Action*, and Maximum Matching, or *Max Matching*. This misspecification setup is motivated by the fact that standard loss minimization guarantees global fidelity but may fail to capture the local signals required for specific downstream optimization tasks (similar to the local biases multicalibration was originally designed to correct in fairness contexts). To model this phenomenon, we generate data using a quadratic ground truth but train a linear predictor to convergence. This ensures that the latter is the "best possible" linear predictor (minimizing MSE) yet remains structurally imperfect—precisely a scenario our procedure is designed to address.

### 5.1. Dataset Construction

In this subsection, we describe how we generate the synthetic dataset for our experiments. In all the experiments, we consider 10-dimensional feature vectors: $\mathcal{X} \subseteq \mathbb{R}^{10}$. We initialize weight matrices $\mathbf{W}_1, \mathbf{W}_2 \in \mathbb{R}^{m \times 10}$ and a bias vector $\mathbf{b} \in \mathbb{R}^m$, where all entries are drawn independently from a standard normal distribution. To create a dataset of $N$ samples, we repeat the following procedure. First, an input vector $x \in \mathbb{R}^{10}$ is drawn from $\mathcal{N}(\mathbf{0}, \mathbf{I})$, where $\mathbf{I}$ is the identity matrix in dimension 10. Then, a raw label vector $\tilde{y} \in \mathbb{R}^m$ is computed as:

$$\tilde{y} = \mathbf{W}_1 \mathbf{x} + \frac{1}{2} \mathbf{W}_2 (\mathbf{x}^2) + \mathbf{b} + z,$$

where $\mathbf{x}^2$ denotes the element-wise square of $\mathbf{x}$, and $z$[‖] is a noise vector. The target vector $y$ is then obtained by standardizing $\tilde{y}$ (normalizing it to have zero mean and unit variance) and applying a sigmoid function to each component. The base predictor $\gamma$ (that we want to improve with our multicalibration pipeline) is a linear predictor, again with a sigmoid function on top, which is trained with gradient descent on $\approx 10^4$ samples generated as above.

**Finding the best action.** Here, each weight corresponds to one of $m$ actions. We consider the setting where $m \in \{4, 16, 64, 256\}$. The family of policies $\mathcal{C}$ we compare against is constructed by picking the $\arg\max$ among re-scaled predictions. Formally, each rule $c \in \mathcal{C}$ is parameterized by a multiplier vector $\lambda(c) \in [0,1]^m$, so that the generic $c$ chooses the action that maximizes the product between $\lambda_i$ and the predicted weight for action $i$. In the experiments, we consider the discrete grid of multipliers $\Lambda = \{0, 0.25, 0.5, 0.75, 1\}^m$. The final set of weight vectors used to define $\mathcal{C}$ is obtained as follows: if $|\Lambda| \leq 1024$, we take the whole $\Lambda$ as the set of multipliers (i.e. we consider all possible combinations); else we sample 1024 vectors uniformly at random from $\Lambda$. When doing this, we disregard all those combinations that induce the same decision rule,

---

[§]A family $\mathcal{F}$ of subsets is called a matroid if (i) $\in \mathcal{F}$, (ii) if $A \in \mathcal{F}$ and $B \subseteq A$, then $B \in \mathcal{F}$, and (iii) if $A, B \in \mathcal{F}$ with $|A| > |B|$, then $\exists a \in A$ such that $B \cup \{a\} \in \mathcal{F}$. In particular, maximizing a linear function with matroid constraints can be done efficiently using the greedy algorithm (Schrijver, 2003).

[¶]The cardinality of any set that is maximal with respect to the matroid property has a fixed cardinality that is called the rank.

[‖]$z$ is drawn from a normal distribution with standard deviation 0.1.

i.e. scalar multiples of a given multiplier vector.

**Maximum matching on graphs.** Here each weight corresponds to an edge in a complete graph, where the number of edges is $m = 45$ (corresponding to the complete graph on $n = 10$ nodes). The family of policies $\mathcal{C}$ is constructed similarly to the previous case, with the difference that the adjusted weights (multipliers times predictions) are fed into a max-weight matching algorithm.

### 5.2. Implementation Details

We analyze the performance according to three metrics, which are Monte Carlo-estimated via 4000 samples:

1. The gap between the performance of the $\arg\max$ onto $\hat{\gamma}$ and that of the best $c \in \mathcal{C}$ onto the initial predictor $\gamma$, i.e. the following difference, the *Utility Gap*:

$$\mathbb{E}\left[\sum_{e \in M_{c^\star}(\hat{\gamma}(x))} y_e\right] - \max_{c \in \mathcal{C}} \mathbb{E}\left[\sum_{e \in M_c(\gamma(x))} y_e\right]$$

2. the difference between the MSE of the initial predictor and the final one (normalized by the number of entries)

3. the improvement in the performance of $M_{c^\star}(\cdot)$, when applied on $\gamma$ or $\hat{\gamma}$.

For a fixed output dimensionality and weight set $\{\mathbf{W}_1, \mathbf{W}_2, \mathbf{b}\}$, we analyze the effects of varying the sample size and target precision $\varepsilon$ within a call to the CHECK routine, which is implemented by consuming disjoint parts of the dataset at each of the 1024 iterations.

We stress that every time we change the dimensionality $m$, we draw again $\mathbf{W}_1, \mathbf{W}_2$ and $\mathbf{b}$, so our results for *Best Action* do not rely on a specific draw of the weights. Since for matching we fix $m$, we average the results over 5 seeds.

### 5.3. Results

In Figure 2, we show the results for both experimental settings. The code for this evaluation is available on GitHub: github.com/facebookresearch/multicalibration_for_matching. Once the CHECK routine yields reliable estimates, the procedure rapidly improves upon the original predictor and the *Utility Gap* vanishes, provided convergence is achieved.

In the *Best Action* setting, a smaller number of arms is associated with a lower initial mean squared error. Moreover, the $\Lambda$ grid is more representative and updates are less sparse, facilitating faster convergence. In all the settings, convergence is further accelerated by moderately high values of $\varepsilon$, which induce more aggressive updates [**] compared to

---

[**]Recall that the learning rate of the gradient descent step is proportional to the precision.

exponentially lower $\varepsilon$ regimes. This dynamic is further detailed in the evolution of MSE during the multicalibration procedure (see Appendix A). In general, for low $\varepsilon$, the update dynamics are smoother, convergence takes longer and a small sample size is often insufficient for such fine-grained errors[††].

It should be noted that the empirical results are more favorable than the worst-case theory in two respects. First, when convergence is at least partially attained, the utility gap becomes positive, not just close to zero from the negative side as the theory guarantees, meaning that the updated predictor can induce a decision rule that outperforms the best rule in $\mathcal{C}$ applied to the original predictor. Second, the sample complexity observed in our experiments is substantially lower than the worst-case one we have derived with strict probabilistic guarantees and when disregarding assumptions on the initial mean squared error. For example, for *Max Matching*, $\varepsilon = 1/4$ reaches positive utility gap when the number of samples consumed is $2^8 \cdot 1024$, while the theoretical worst-case total sample complexity to achieve such a result is at least $10^5 \cdot 2^8 \cdot \log(1024)$, which is hundreds of times higher. Similarly, for the same setting, $\varepsilon = 1/8$ reaches positive utility gap when we consume $1024 \cdot 2^{10}$ samples, but the theoretical guarantee would be at least $10^5 \cdot 2^{12} \cdot \log(1024)$, which is thousands of times higher.

As stressed in our analysis in terms of mean squared error convergence, one of the relevant factors here is a reduced number of iterations to get a contained utility gap. As an example, consider the *Best Action* setting with $m = 256$, $\varepsilon = 1/16$, using $2^{10}$ samples in each call to CHECK. For this setup, the routine achieves convergence and the required utility gap guarantee with 1024 iterations, but the worst-case theoretical bound on the number of iterations is $4 \cdot 256 \cdot 2^8$, which is approximately 260 times as much.

Notably, noisy estimates from CHECK may partly explain the curve in Figure 2b for the highest $\varepsilon$, in the *Max Matching* setting. High variance in the estimation of small multicalibration errors can increase the chance of detecting violations, triggering updates even when the real signal is weak. However, once the CHECK sample size is sufficiently large, this effect diminishes, and the observed behavior is governed by genuine violations rather than estimation noise.

## 6. Limitations

While the theoretical analysis in Section 3 provides provable guarantees for our post-processing pipeline, its bounds are subject to certain statistical and computational limitations.

---

[††]Recall that for matching the threshold for CHECK violations is further shrunk down, since the considered $\alpha$ is $\varepsilon$ scaled down by the matching size. For *Best Action*, we instead do not need to scale, as explained in Section 4.

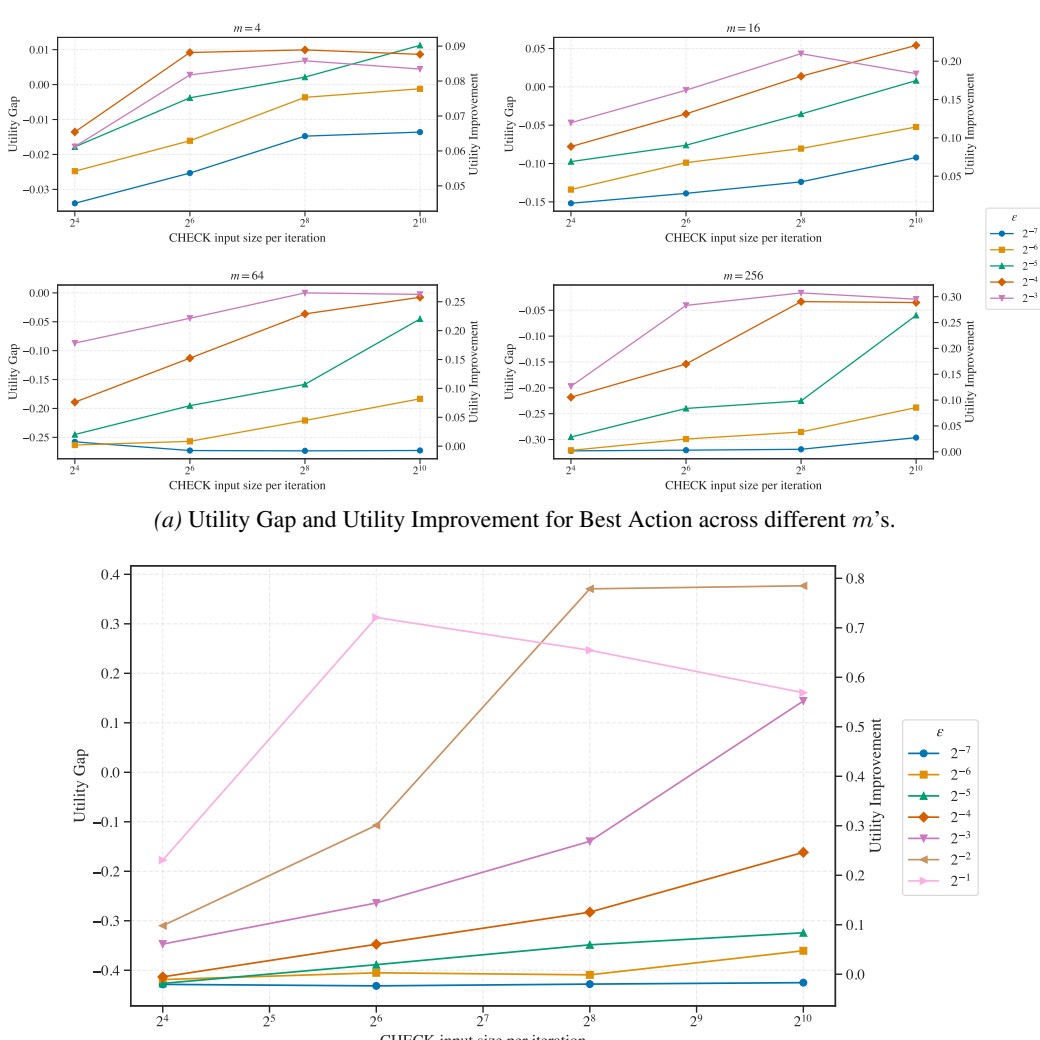

*(a)* Utility Gap and Utility Improvement for Best Action across different $m$'s.

*(b)* Gap and Improvement for Max Matching, with $m = 45$ and averaging over seeds.

*Figure 2.* **Utility Gap and Utility Improvements** Improvement and Gap are a constant amount away from each other, so we plot them together and have one vertical axes for each. Improvement is always positive and the gap shrinks to 0 or becomes positive when $\varepsilon$ is not too low and the algorithm converges. The x-axis, for the number of samples given to CHECK at every iteration, is in log-scale (base 2).

From an algorithm design perspective, the obtained guarantee is w.r.t. to a class of matching algorithms $\mathcal{C}$ decided in advance. Statistically, the sample complexity exhibits a polynomial dependence on $n$ and a $1/\varepsilon^4$ dependence, with this possibly improved to $1/\varepsilon^3$ using more computationally intensive differential privacy techniques, as we wrote in Section 3. Computationally, the number of boosting iterations can be high in the worst case (i.e. $r = m$), scaling cubically in $n$ and quadratically in $1/\varepsilon$. The requirement to enumerate over $\mathcal{W}$ and compute matching algorithms for each sample may also pose a bottleneck without parallelization.

Nonetheless, as highlighted in our experimental section, these results are frequently overconservative, and practical performance can exceed these worst-case limits. For instance, our proofs show that a low initial MSE $r$ or the identification of larger than required violations by CHECK can significantly reduce iteration counts and running time, and thus sample complexity.

## Impact Statement

This paper presents work whose goal is to advance the field of Machine Learning. There are many potential societal consequences of our work, none of which we feel must be specifically highlighted here.

## Acknowledgments

The work of SDG, SF, FF, SL, and CS is partially supported by the Meta/Sapienza project on "Online Constrained Optimization and Multi-Calibration in Algorithm and Mech-

anism Design". A significant part of this work was done while MR was at Sapienza University of Rome.

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

## A. Convergence and Mean Squared Error plots

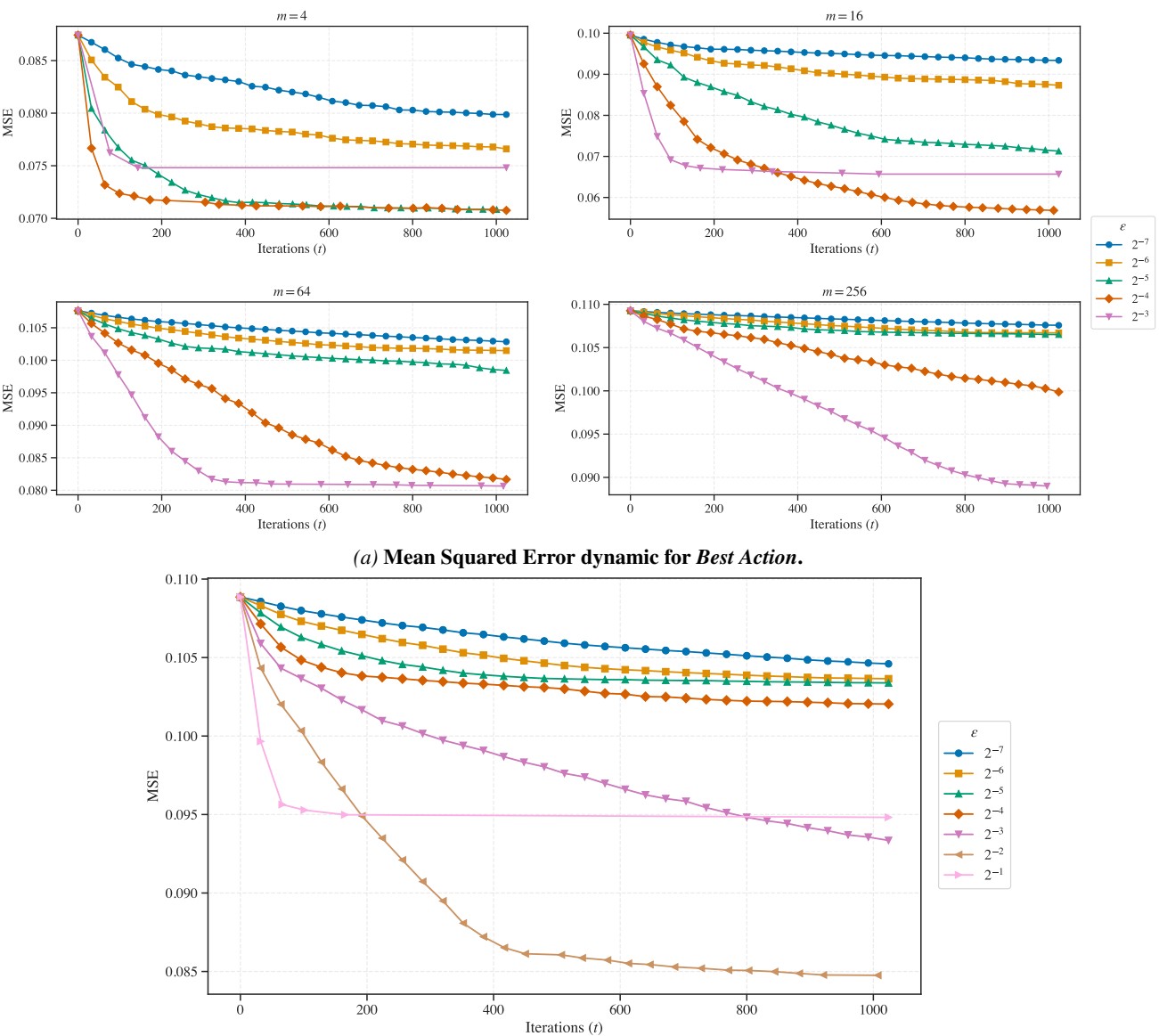

*(a)* **Mean Squared Error dynamic for *Best Action*.**

*(b)* **Mean Squared Error dynamic for *Maximum Matching*,** for a single seed.

*Figure 3.* **Mean Squared Error for the *Best Action* setting and the *Maximum Matching* setting.** In this plot, we set the number of samples to CHECK to be 1024, so that we can focus on the MSE dynamic given that the averages are moderately well estimated for most of the $\varepsilon$ configurations. The number of iterations is capped to 1024, as explained in the main body. Convergence is easier when $m$ is low and $\varepsilon$ is moderately high because the update is less sparse, and the optimization is more aggressive. However, if $\varepsilon$ is too high, then the optimization stops very early, since every time a fresh sample is drawn, no new violations are found. Recall that here we are not early exiting the procedure when this happens: we continue to draw and use CHECK until the iteration cap is reached.

