# OpenReview forum: "Multicalibration Yields Better Matchings"
_ICML.cc/2026/Conference — ICML 2026 regular_

### Official Review · Reviewer_6yuD · 2026-03-11

**Soundness:** 3
**Presentation:** 2
**Significance:** 2
**Originality:** 3
**Overall Recommendation:** 3
**Confidence:** 2

**Summary:**

The paper studies the classic matching problem in stochastic graphs, in which the edge weights are outcomes of a random process. It assumes access to a black-box predictor $\gamma(e)$ that returns an estimate for the weight $y_e$ of any given edge $e$.

The paper's goal is to use so-called multicalibration to obtain a predictor $\hat{\gamma}$ with the following property: If we use $\hat{\gamma}$ with an exact matching algorithm, then this yields almost (up to $\varepsilon$ additive error) the same matching size as when using the best-possible decision rule (algorithm) when using $\gamma$ directly. This is interesting because the black-box predictor $\gamma$ may have bias, such that one may want to use a "specialized" (non-optimal) matching algorithm that is tailored towards the bias in $\gamma$.

The paper's main contribution is to show such a predictor $\gamma{\hat}$ can be constructed by using an iterative boosting procedure.

**Compliance With Llm Reviewing Policy:**

Affirmed.

**Final Justification:**

After the rebuttal and reading the other reviews, I would like to stick with my initial score. My impression is that the paper would benefit from a revision before reaching the level of a top conference paper.

**Key Questions For Authors:**

Can you give a real-world example in which one has to solve matching in a setting as captured by the submission?

**Limitations:**

Yes.

**Strengths And Weaknesses:**

This is not my research field, so I can only provide an outsider opinion.

I believe the topic studied by the paper is interesting. The area of algorithms that operate with uncertain data/advice has recent a lot of attention recently, and I believe studying the classic matching problem in such a setting is interesting. As an outsider, I find the obtained results interesting (but I have to stress that I do not know the literature in this area and cannot judge how novel or surprising they are).

Regarding the downsides of the paper, I can say that as a non-expert it is quite hard to read and would benefit from further explanations (see my comments to the authors below). The paper would also benefit from additional motivation or applications of the studied problems.

Overall, I have no strong opinion on this paper.

Comments for the authors:
- For a non-expert, the example in the introduction (the paragraph starting with "For example, ...") is too compressed to be understandable.
- Line 168: I was briefly confused what "optimal" refers to here (e.g., Edmond's algorithm is not optimal w.r.t. running time). Perhaps it would be less ambiguous to say an "exact" algorithm.
- I find it confusing that $\mathcal{C}$ is never formally introduced in Sections 2 or 3. It is only briefly mentioned in the introduction that $\mathcal{C}$ is a "class of decision rules" but later it is basically used as a set of possible matching algorithms. I think this should be mentioned somewhere explicitly.
- Equation (3): There is a parenthesis missing in the subscript of the first line. It would be beneficial to explain what $p$ is.
- It would be beneficial if the paragraph ("We claim ...") that appears before Theorem 3.1 would appear after Equation (2).
- Given that in Section 3.2 it is stated that the CHECK function is a "core component", I was surprised that its implementation is only given within the proof of Theorem 3.3.
- Figure 2: The axis labels are much too small.

---

> ### Author Rebuttal · Authors · 2026-03-31
>
> We thank the reviewer for their valuable feedback and will revise the paper to address the clarity concerns raised.
>
> ‘For a non-expert, the example in the introduction (the paragraph starting with "For example, ...") is too compressed to be understandable.’
>
> For clarity, we first elaborate on the example provided in the Introduction; we will incorporate a refined version of this explanation into the final manuscript. By definition, the predictor evaluates the first arm perfectly, resulting in a bias of zero.
> We now focus on the second arm, whose true value is $\frac{1}{\varepsilon}$. With probability $1-\varepsilon$, the predictor outputs zero, and with probability $\varepsilon$, it outputs $\frac{1}{\varepsilon^2}$. Consequently, the expected value of the predictor for the second arm is exactly $\frac{1}{\varepsilon}$. Because this matches the true value, the bias for the second arm is also zero. Observe that the second arm strictly dominates the first, particularly for small values of $\varepsilon$. However, selecting the argmax based on the predictor's values results in choosing the suboptimal first arm with probability $1-\varepsilon$. Thus, as $\varepsilon \rightarrow 0$, the gap in utility between the two arms grows arbitrarily large, while the probability of making the suboptimal decision using the standard argmax policy approaches 1.
>
> We now address their key question. A natural application arises in load balancing over transportation networks; see _Data Analysis and Optimization for (Citi) Bike Sharing_ (AAAI, Brian O'Mahony and David B. Shmoys, 2015). In Section 4, the authors introduce the “Mid-Rush Pairing” problem, where stations with negative bike flow are matched to stations with positive flow, with edge weights determined by distances between stations. This matching is then used to rebalance bike availability during rush hours.
> Their approach is purely optimization-based and incorporates additional operational constraints. However, if one replaces the edge weights with predictions that account not only for distance but also for anticipated station load—learned from historical data under a stationarity assumption—this setting naturally aligns with our framework.

---

> > ### Author Rebuttal · Reviewer_6yuD · 2026-04-01
> >
> > Many thanks for your clarifications and for pointing out the application. I will consider increasing my score after deliberating with the other reviewers.

---

### Official Review · Reviewer_qFxA · 2026-03-12

**Soundness:** 3
**Presentation:** 3
**Significance:** 3
**Originality:** 3
**Overall Recommendation:** 4
**Confidence:** 4

**Summary:**

This paper addresses a practical problem in optimization: when we need to find the best matching in a weighted graph but only have access to imperfect predictions of the actual weights (rather than the true weights themselves). In real world systems, machine learning models predict these weights based on available context, but these predictions are never perfect. The standard approach is to simply compute the optimal matching using the predicted weights, but this can perform poorly when predictions are biased in certain ways. The authors propose using multicalibration a fairness concept that requires predictions to be unbiased across many different subgroups as a way to fix imperfect predictors so that the standard optimization approach works well again.

The main contribution is showing how to transform any given predictor into a multicalibrated version that has a useful guarantee: running the standard max weight matching algorithm on the corrected predictions will perform at least as well as the best decision rule from a predefined class applied to the original predictions. The authors focus on the concept of using multicalibration as a post-processing step that improves predictors for downstream optimization tasks. The authors present a critical problem in real-world matching systems where engineers often resort to manual patches and heuristics when their predict-then-optimize pipeline fails in certain scenarios. The paper provides theoretical sample complexity bounds showing how many data points are needed to achieve this calibration, and includes experiments on synthetic data demonstrating that the approach works in practice, the utility gap between the corrected predictor and the best alternative decision rule shrinks to zero as expected.

**Compliance With Llm Reviewing Policy:**

Affirmed.

**Final Justification:**

Will proceed with initial evaluation

**Key Questions For Authors:**

1- How does the method perform on real-world data?


2- Can you provide a direct empirical comparison with the baseline approach of estimating each decision rule's performance separately?
You mention this alternative requires fewer samples but lacks the modularity benefits of your approach. A head-to-head comparison showing the tradeoff between sample efficiency and downstream flexibility would help readers understand when to prefer your method.

3-  What is the actual sample requirement in your experiments, and how does it compare to your theoretical bounds?
The worst-case bounds are quite pessimistic. It would be useful to know how tight these bounds are in practice. If the actual sample requirements are smaller than the theory suggests, this would make the method more appealing for real applications.

**Limitations:**

The paper would benefit from a dedicated limitations section discussing the practical constraints of the method, such as the high sample complexity, the need to specify the class C in advance, and the computational cost of the iterative boosting procedure.

**Strengths And Weaknesses:**

Strengths
The paper addresses a real problem that occurs in industry systems: when you have a predictor feeding into an optimization routine, the standard "predict-then-optimize" approach can fail even with unbiased predictors. The example in the introduction with two arms clearly illustrates why this matters.
 A key practical advantage is that once the predictor is multicalibrated, you only need to run a single standard algorithm (max-weight matching) at test time. This is simpler than selecting among multiple decision rules and avoids accumulating technical debt from manual patches.
 The paper shows how the same ideas extend to other linear optimization problems like best-action selection, learning with rejection, and matroid constraints. This increases the usefulness of the contribution.

Weaknesses
 The experiments are only on synthetic data with a specific type of model misspecification (quadratic ground truth with linear predictor). It remains unclear how well the method works on real-world matching problems with natural distribution shifts or more complex forms of predictor error.
The worst-case bound of O(n⁵/ε⁴) is quite large for graphs with many nodes. While the refined analysis helps when the initial predictor is good, the paper does not provide empirical evidence about how many samples are actually needed in realistic scenarios.

---

> ### Author Rebuttal · Authors · 2026-03-31
>
> We thank the reviewer for their constructive feedback. As suggested, we will include a dedicated limitations section in the revised manuscript. Below, we answer the reviewer's specific questions, which should also address the concerns raised in the Weaknesses section.
>
> Questions:
> 1. As a proof of concept, we just run the experiments on a synthetic data generation process. We agree it is an interesting future direction to investigate the performance of our approach on real-world data.
> 2. We provide the required comparison for our synthetic setup in Figure 2 in the repo [at this link](https://anonymous.4open.science/r/rebuttal-plots-065A/README.md), for a specific configuration, but similar results arise with other setups. In the experiment we consider 105 edges, $\varepsilon = 0.05$ (which is then scaled down by the matching size), 256 sampled decision functions, 512 maximum iterations and 64 as batch size. The same amount of samples are then accumulated at every iteration by the baseline method, which then updates the averages and thus its choice of the best decision function. As expected, the baseline quickly converges to the best in the class, while the multicalibrated predictor at convergence gets closer to the best in the class, without reaching its performance. Notice that this happens mainly because $\varepsilon$ is in this case quite large.
> 3. To address this question, we first refer to Figure 2(b). Here, most $\varepsilon$ configurations achieve a positive utility gap using a very small number of samples per CHECK iteration—significantly fewer than assumed in the theoretical analysis. To make this concrete, consider $\varepsilon = 2^{-6}$ (similar reasoning applies to the other curves). In this case, the utility gap becomes positive with only 256 samples per CHECK iteration. Given 1024 boosting iterations, this corresponds to approximately $2.5 \times 10^4$ total samples. By contrast, the worst-case theoretical regime demands at least $10^5 \cdot 2^{24} = 10^{12}$ samples, highlighting a substantial gap between theory and practice.
> This gap stems from more than just the algorithm's practical robustness to moderately noisy mean estimates (assuming no early stopping). It is primarily driven by the fact that the desired utility gap guarantees are achieved in far fewer iterations than the worst-case analysis suggests. For example, in Figure 2(a) (bottom-right plot), which considers the simpler value maximization setting, the $2^{10}$ sample size for $\varepsilon = 2^{-4}$ is clearly sufficient to estimate the relevant averages accurately. In this setting, the algorithm attains the $2^{-4}$ guarantee using roughly 16 times fewer samples than theoretically predicted, not accounting for log-factors, which would increase the difference. Because the estimation error is already small in this regime, this improvement is attributable entirely to the reduced iteration count.
> Regarding the relationship between the number of iterations and the utility gap guarantee, we refer to Figure 1 in the repo [at this link](https://anonymous.4open.science/r/rebuttal-plots-065A/README.md): the configuration is the same as the one for Figure 2. Here, we disregard the estimation error in each iteration and only focus on the error as a function of the number of iterations, the worst case theoretical one against the empirical one. As it can be seen, the difference is relevant, especially with a relatively low initial mean squared error.

---

> > ### Author Rebuttal · Reviewer_qFxA · 2026-04-02
> >
> > Would be good to see additional experiments.

---

### Official Review · Reviewer_BkFN · 2026-03-13

**Soundness:** 4
**Presentation:** 2
**Significance:** 3
**Originality:** 3
**Overall Recommendation:** 4
**Confidence:** 3

**Summary:**

The paper considers the problem of max-weight matching where true edge weights are unknown and we only have access to potentially erroneous predictions of them. Traditionally, one would utilize these predictions by simply finding the max-weight matching based on the predicted weights. The problem is that even if these estimates are unbiased, i.e., accurate on average, they might still lead to far from optimal solutions, since unbiasedness alone might not be enough. To get around this problem, one might use different heuristics for different situations . In order to give an all-in-one solution to this problem, the authors bring the idea of multicalibration from the algorithmic fairness literature to this setting. The idea is to adjust the prediction before optimization so that running an exact max-weight matching algorithm on the multicalibrated prediction performs, in expectation, almost as well up to an additive $\epsilon$ as the best decision rule from a class $\mathcal{C}$ applied to the original predictor. The authors prove that such a multicalibrated predictor exists. The max-weight matching problem is used as a running example in the paper, but the authors also argue that the results extend more broadly to linear maximization tasks with deterministic constraints, with suitable adjustments. The paper also includes sample complexity bounds and experiments on synthetic data.

**Compliance With Llm Reviewing Policy:**

Affirmed.

**Final Justification:**

My final recommendation is weak accept. The rebuttal reinforced my prior assessment. I do believe the paper has novelty and I am partially convinced about its practicality; however, the technical contribution still seems relatively thin.

**Key Questions For Authors:**

I do not have any questions for the authors.

**Limitations:**

yes

**Strengths And Weaknesses:**

**Soundness**

The paper appears technically sound. The main claims are supported by rigorous theoretical analysis as well as experiments.

**Presentation**

The paper is somewhat difficult to follow. The main ideas are quite conceptual, and more intuition or concrete examples would make the results easier to absorb.

**Significance**

The paper offers a clean and principled way to replace heuristic fixes in predict-then-optimize settings, which is a meaningful contribution. However, the practical impact of the results is somewhat unclear to me, especially since the experiments are only on synthetic data.

**Originality**

The paper is original in the way it combines ideas from algorithmic fairness with combinatorial optimization. This connection leads to a new perspective on how to improve decision quality using calibrated predictions. However, the contributions seem more conceptual than technical and the proofs largely rely on existing techniques.

---

> ### Author Rebuttal · Authors · 2026-03-31
>
> We thank the reviewer for their valuable feedback. We will improve the clarity in terms of presentation and add examples in the camera-ready.

---

> > ### Author Rebuttal · Reviewer_BkFN · 2026-04-02
> >
> > I do not have any further concerns.

---

### Official Review · Reviewer_e2WA · 2026-03-21

**Soundness:** 3
**Presentation:** 2
**Significance:** 2
**Originality:** 2
**Overall Recommendation:** 3
**Confidence:** 3

**Summary:**

This paper has pointed out the critical problem that modern ML pipelines of (prediction -> algorithmic decision making -> adhoc patches to make up for prediction error) often break due to predictive error. Also, this can happen even when the predictions are unbiased in expectation. Instead of adding hard-coded heuristics to patch the decision logic, as is the traditional approach used by practitioners, the authors propose a rigorous, data driven alternative of multicalibration techniques.  By defining a "filter" class $\mathcal{W}$ that captures the behavior of different individuals in a finite algorithm set $\mathcal{C}$, the proposed algorithm post-process the predicted edge weights to guarantee that it is almost unbiased in a "brute-force" way (on every choice that the algorithms can make). The authors provided both theoretical and empirical guarantees for this approach.

**Compliance With Llm Reviewing Policy:**

Affirmed.

**Key Questions For Authors:**

See above.

**Limitations:**

Yes

**Strengths And Weaknesses:**

Strengths:
- Conceptual novelty. The problem studied is highly motivated, and in fact often encountered in real industry. The problem studied, max-weight matching, as the authors have pointed out, also covers many real applications. While this paper is no pioneer in inventing the multicalibration - optimization framework, based on my search, this idea of using it in algorithm engineering is novel (whether that is simply a twist of existing ideas or genuinely original concept is up to discussion).
- The multicalibration approach belongs to a theoretically elegant framework.
- The proposed pipeline successfully isolates the "fix" to the data layer; once the multicalibration is done downstream algorithms could be run rather independently.

Weaknesses:
- The experiment designs are less convincing. The scale of the datasets doesn't justify its performance in large-scale systems. The policy class seems to be rigid and a poor proxy for what people do in practice.
- I wouldn't consider the running time small since it is even more than $n$ cube. This puts a question mark on how practical this method is.
- Lack of algorithm transparency. The comparison to related works, especially Gopalan et al. (2022b), should be described in more technical details. As is, it is hard to tell how much originality the algorithms can claim in Sec 3. The main algorithm 1, especially the CHECK function, is a variant of projected gradient descent, if I'm not mistaken; and so is the proof of performance. This connection should have been made clearly in the paper.

Minor issues:
- in line 82 the $n$ in running time bound appears for the first time but hasn't been defined before.
- in line 124 right column "account for for low sample frequency"; there is a redundant "for".
- Def 2.2 is phrased poorly. The authors should have clarified that $\mathcal{W}$ sort of plays a role like auditors acting as a filter that only looks at the errors for some specific edges under some specific conditions.

---

> ### Author Rebuttal · Authors · 2026-03-31
>
> We thank the reviewer for their valuable feedback and comments. We will correct the typos and minor imprecisions pointed out in the revised version of the paper. Below, we address the points raised in the Weaknesses section, in order.
>
> “The policy class seems to be rigid and a poor proxy for what people do in practice.”
>
> In real-world scenarios, we expect a decision-maker to have prior knowledge of the problem and to tailor specific decision functions accordingly. In our synthetic scenario, however, we aimed to design a policy class that remains competitive and meaningful without any a priori knowledge. Indeed, the functions within this class reweight the predictions to simulate a 'brute-force' approach across every possible decision. We consider this a natural strategy for this setting, as the reweighting may compensate for some bias intrinsic in the original estimator.
>
> “I wouldn't consider the running time small since it is even more than cube.”
>
> We acknowledge that our sample complexity is higher than standard PAC learning rates (i.e. O($\frac{1}{\alpha^2}$)), which consequently affects the algorithm’s running time. However, prior work suggests an inherent separation between standard learning and multicalibration with respect to the parameter $\alpha$: the best known methods achieve rates of $\frac{1}{\alpha^3}$ using adaptive data analysis and more complex, computationally intensive procedures (as noted on page 6), while more standard and lightweight approaches attain $\frac{1}{\alpha^4}$. We refer to the discussion following Proposition 34 in Gopalan et al. (2022b) for further details. Notably, a worse dependence on $\alpha$ also translates into a worse dependence on the problem scale $n$.
>
> From a technical perspective, our presented bound is a consequence of the potential-based analysis, which assumes worst-case violations on the order of $O(\alpha)$ at each step. In practice, however, detected violations are typically significantly larger, leading to a substantially smaller number of iterations required to achieve the target guarantees. We further emphasize two practical implications of our theoretical bounds. First, while Theorem 3.1 dictates that $(\mathcal{W}, \alpha(\varepsilon,m))$-MC is sufficient to guarantee a Utility Gap of at most $\varepsilon$, our empirical results (Figures 2(b) and 3(b)) show that a small utility gap can be realized well before full multicalibration is achieved. Second, Theorem 3.3 shows that a smaller initial MSE $r$ proportionally decreases the overall iteration complexity.
>
> We point to Figure 1 in the repo [at this link](https://anonymous.4open.science/r/rebuttal-plots-065A/README.md) for a more detailed exploration of the behavior of the utility gap when matched against the number of iterations. We made a test — under our same synthetic setup — with 105 edges, $\varepsilon = 0.05$ (which is then scaled down by the matching size), 256 sampled decision functions, 512 maximum iterations and 64 as batch size. We then disregarded the estimation error in each iteration and only focused on the error as a function of the number of iterations, the worst case theoretical one against the empirical one. As it can be seen, the difference is relevant, especially with a relatively low initial mean squared error.
>
> Finally, our graph experiments show that far fewer samples are needed in practice at each iteration than prescribed by Theorem 3.3. This is partly due to the conservative nature of the concentration bounds utilized in the proof. Combined with the aforementioned points regarding the iteration count, we expect the practical running time and sample complexity of the algorithm to be substantially lower than our theoretical bounds suggest.
>
> “Lack of algorithm transparency”
>
> As noted in lines 221-232 of the manuscript, our algorithm is an adaptation of the one proposed by Gopalan et al. (2022b). We stress that the framework underlying this reference builds on prior work; indeed, most (if not all) algorithms in the multicalibration literature adapt the foundational procedure introduced by Hebert-Johnson et al. (2018)—namely, iterative boosting procedures that use a weak learner to identify violations of the multicalibration condition before performing a projected gradient descent step. However, adapting this framework to our setting required some care in the analytical steps and in the description of the algorithm. First, we consider a more flexible notion of multicalibration, and we focus on learning vectors in $[0,1]^d$ rather than standard multi-class classification (which is the standard approach and entails projecting onto the simplex). Second, our analysis explicitly extracts the dependence on the initial MSE, $r$, which is a meaningful parameter for the post-processing methodology we introduce.

---

> > ### Author Rebuttal · Reviewer_e2WA · 2026-04-06
> >
> > I have read the authors' acknowledgment and other reviewers' feedback. The rebuttal have provided more details, however it doesn't change my general feeling about the algorithm designs and practicality. I have decided to keep my score after some deliberation.

---

### Decision · Program_Chairs · 2026-04-30

**Decision:**

Accept (regular)

**Comment:**

The reviewers appreciated the strong motivation of the problem, the conceptual novelty, and the elegance of the used framework. The paper appears technically sound, and the main claims are supported by rigorous theoretical analysis as well as experiments. The most appreciated contribution is the conceptual connection - a new perspective that comes from combining ideas from algorithmic fairness with combinatorial optimization. It makes for a good contribution to ICML.

Improvements are, however, possible, and we hope that they will be made for the final version:
- The practicality was a subject of rebuttals and debate and the reviewers feel partially convinced. Concerns were raised about the need to specify the class in advance and about the running time / sample complexity. The latter does seem to be better in practice than the worst-case theoretical bounds would suggest (as the authors also argued in the rebuttal). However the experiments are somewhat limited, small in scale, and based on synthetic data. Higher-quality empirical evidence, based on more real-world scenarios (e.g. with natural distribution shifts or more complex forms of predictor error) would be desirable.
- It was felt that the paper would benefit from a dedicated limitations section discussing the practical constraints of the method, as well as more illustrative examples and intuition. Another criticism made was that the technical contribution is somewhat thin compared to prior work, and that the contribution was primarily conceptual. Thus a more comprehensive comparison to prior work, especially Gopalan et al. (2022b), would also bolster the manuscript.